# Efficient Ensemble Classification for Multi-Label Data Streams with Concept Drift

**Yange Sun [1,2,3,*], Han Shao [1] and Shasha Wang [1]**

[1]  School of Computer and Information Technology, Xinyang Normal University, Xinyang 464000, China; shao0102@xynu.edu.cn (H.S.); wss1020@126.com (S.W.)
[2]  School of Computer and Information Technology, Beijing Jiaotong University, Beijing 100044, China
[3]  Henan Key Lab of Analysis and Applications of Education Big Data, Xinyang Normal University, Xinyang 464000, China
*  Correspondence: ygsun1982@126.com; Tel.: +86-188-3769-1652

**Abstract:** Most existing multi-label data streams classification methods focus on extending single-label streams classification approaches to multi-label cases, without considering the special characteristics of multi-label stream data, such as label dependency, concept drift, and recurrent concepts. Motivated by these challenges, we devise an efficient ensemble paradigm for multi-label data streams classification. The algorithm deploys a novel change detection based on Jensen–Shannon divergence to identify different kinds of concept drift in data streams. Moreover, our method tries to consider label dependency by pruning away infrequent label combinations to enhance classification performance. Empirical results on both synthetic and real-world datasets have demonstrated its effectiveness.

**Keywords:** data streams; multi-label; concept drift; ensemble classification; label dependency

## 1. Introduction

In recent years, sensor networks [1], spam filtering [2], intrusion detection [3], and credit card fraud detection [4] have contributed to different new applications in continuously arriving data known as data streams [5]. In the data streams model, instances arrive at a higher rate, and the algorithms must process them with strict constraints of time and memory [6].

However, traditional methods focus on classifying data streams under single-label scenarios where each instance belongs to a single label. In practice, many real-world applications involve data with multi-label data streams. A multi-label data stream is a kind of stream that shares the same properties as multi-label data [7]. Typical multi-label data include news articles, e-mails, RSS feeds, medical text, etc. Multi-label stream classification is a non-trivial task, because traditional multi-label classification approaches work under the batch settings. An important feature of multi-label data streams is concept drift, i.e., the underlying distribution of data may change over time. Such changes might deteriorate the predictive accuracy of classifiers.

In brief, concept drifts can be categorized depending on speed into gradual and abrupt concept drifts [8]. Several approaches have been presented to handle concept drifts, which fall into three general groups, i.e., window-based approaches, weight-based approaches, and ensemble-based approaches [9]. Among them, ensemble classifier is particularly effective and popular, but it calls for generalization to accommodate multi-label data streams environments.

Most existing methods can only deal with one type of concept drift. However, in the real world, data stream may contain different types of concept drifts. Therefore, it is highly crucial that an effective classifier must be able to respond to both sudden and gradual changes.

It is common in real-world data streams for previously seen concepts may reappear in the future. For example, weather forecasting models change with the seasons, and a popular topic may appear in a social network at a particular time of the year. This demonstrates a unique kind of drift known as recurring concepts. However, only few approaches take it into consideration recurring concepts in the context of multi-label data streams.

One important advantage of ensemble techniques in streaming data is that they can handle recurring concepts. Since ensembles contain models built from different parts of the data streams, such models can be reused to classify new instances if they have come from the same concept. Another advantage is the ability to utilize traditional classification algorithms in the concept drift scenarios. Following these critical motivations, we devise an efficient scheme named Multi-Label ensemble with Adaptive Windowing (MLAW) for multi-label data streams classification. The ensemble maintains a weighted set of component classifiers and predicts the class of incoming instances by aggregating the predictions of components using a weighted voting mechanism [10]. In summary, the contribution of this paper is a simple and effective method for multi-label data stream scenarios. The key contribution of our algorithm is threefold:

(1) To address the issue of concept drift, a change detection mechanism is employed to offer fast reactions to sudden concept changes. Moreover, a periodic weighting mechanism is adopted to cope with the gradual concept drift.

(2) To deal with the problem of recurrent concept drift, the Jensen–Shannon divergence is selected as a metric to measure the distribution between two consecutive windows, which represent the older and the more recent instances, respectively. Moreover, a novel ensemble classification framework is presented, which maintains a pool of classifiers (each classifier represents one of the existing concepts) and predicts the class of incoming instances using a weighted majority voting rule. Once a change occurs, a new classifier is learned and then a new concept is identified and added to the pool.

(3) A label dependency is taken into account by pruning away some infrequent label combinations to enhance classification performance.

The rest of this paper is arranged as follows. In Section 2, we discuss existing work on multi-label stream classification. The details of proposed algorithm are elaborated in Section 3. Experimental results and discussions are shown in Section 4. Finally, Section 5 comes to the conclusion.

## 2. Related Work

### 2.1. Basic Concepts and Notations

Let $\chi$ denote the input feature space, $\chi \subset R^d$, and $\mathcal{L} = \{1, \dots, L\}$ is a set of $L$ possible labels. Let $(x_t, y_t)$ be the instance at time point $t$ in the data streams, where $x_t = \left(x_t^1, \dots, x_t^d\right) \in \chi$ is assigned with a set of labels $y_t = (y_t^1, \dots, y_t^L) \in \{0, 1\}^L$, here $y_i^j = 1$ if the $j$th label is relevant (otherwise $y_i^j = 0$). A theoretically infinite stream of multi-label instances is denoted as $S = \{(x_1, y_1), \dots, (x_n, y_n), \dots\}$. The aim of multi-label stream classification is to train a classifier based on both historical and current instances in the stream to predict the label sets of incoming instances.

Next, some related definitions will be introduced.

**Definition 1 (Concept Drift).** *The term concept drift is defined as the joint distribution of the data evolving over time, i.e., $P_t(x_i, y_i) \neq P_{t+1}(x_i, y_i)$ [9–11].*

**Definition 2 (Sudden Concept Drift).** *As shown in Figure 1a, sudden concept drift means that the original data distribution will be changed directly to a new data distribution at a specified time.*

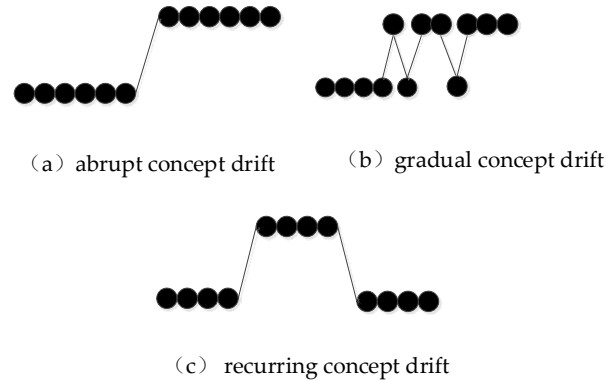

(a) abrupt concept drift　　　(b) gradual concept drift

(c) recurring concept drift

**Figure 1.** Types of concept drift.

**Definition 3 (Gradual Concept Drift).** *As shown in Figure 1b, gradual concept drift means that the probability of the old data distribution will decrease, and the probability of a new distribution will increase during a period of time until the new distribution substitutes the old one.*

**Definition 4 (Recurring Concept Drift).** *A recurring concept drift happens when the records from a period k are generated from the same distribution as a previously observed period $P_k(x_i, y_i) = P_{k-j}(x_i, y_i)$, as shown in Figure 1c.*

**Theorem 1 (Hoeffding Bound).** *With probability $1 - \delta$, the estimated mean after n independent observations of range R will not differ from the true mean by more than $\varepsilon$*

$$\varepsilon = \sqrt{\frac{R^2 \ln\left(\frac{1}{\delta}\right)}{2n}} \tag{1}$$

*where $\delta \in (0, 1)$ is a user defined confidence parameter.*

### 2.2. Multi-Label Learning Algorithms

In the past few years, multi-label learning has attracted much attention during the past few years and many multi-label methods have been proposed [12], and they are mainly divided into two general categories—problem transformation and algorithm adaptation [13,14].

The first group of methods extends specific learning algorithms in order to address multi-label data directly. Examples include decision trees [15], lazy learning [16], neural networks [17], boosting [18], etc. The second group of methods is algorithm independent and they transform the multi-label classification problem into either one or more single-label classification problems. Binary Relevance (BR) and Label Powerset (LP) are two well-known problem transformation methods [14]. According to the BR approach, a multi-label classification problem is decomposed into multiple, independent binary classification problems. The main disadvantage of BR approach is that possible dependencies among the labels are ignored. Conversely, the LP method treats the entire label sets as single labels. Although LP takes label dependency into account directly, when the number of labels increases, the number of possible combinations increases exponentially. Additionally, LP can only predict label sets observed in the training data. One of the main drawbacks of this approach is that each combination may be associated with very few training instances.

Existing methods mainly address this issue via training a prediction model for each label based on the combination of original features and the labels on which it depends. Obviously, exploiting such kind of information would lead to more accurate models. A multitude of methods have been put forward, such as Classifier Chains (CC) [19], IBLR-ML [20], LEAD [21], and others are summarized in [22] based on the types of label dependency that have been exploited.

## 2.3. Classifiers for Multi-Label Data Streams

There are limited numbers of techniques available in the literature on multi-label classification. The first approach to multi-label stream classification was presented in [23]. This method is batch-incremental, using $k$ batches of instances to train Meta-BR (MBR). To deal with concept drift, every $S$ instances of the oldest model is replaced by a model built on the latest block. Kong and Yu proposed a one-pass approach, named Streaming Multi-lAbel Random Trees (SMART) [24], to efficiently update model structure and statistics on each tree node over the multi-label data streams. But this method does not explicitly change detection mechanism, but rather just phases out older instances over time. Similar idea was presented in [25]. Read et al. proposed a novel method Multi-label Hoeffding Tree (MHT), which was an extension of Hoeffding Tree. The main difference is that, to deal with concept drift, SMART used simple fading functions at the nodes of each tree to simply fade out the influence of past data, rather than MHT using a drift detection mechanism to detect the change. Their subsequent work, named as EaHTps [7], presented a method that was a bagged ADWIN-monitored ensemble of Multi-label Hoeffding Tree classifiers. To deal with class imbalance and multiple concept drift, Spyromitros-Xioufis et al. presented a Multiple Windows (MW) method that employs two windows for each label, one for positive and one for negative instances [26]. However, the windowing mechanism was too sophisticated. A method, which aims to detect concept drift, based on label grouping and entropy for multi-label data was presented in [27]. Recently, Nguyen et al. introduced a Bayesian based multi-label method for multi-label data stream classification [28]. The method can deal with missing values and concept drifts simultaneously. Roseberry et al. presented a Multi-Label classifier using a Self-Adjusting Memory (ML-SAM-$k$NN) for data streams [29]. In [30], a dynamically-weighted stacked ensemble, called GOOWE-ML that used spatial modeling to assign optimal weights of classifiers was designed.

The aforementioned approaches try to combine existing data streams and multi-label classification methods. Nevertheless, they fail to deal explicitly with the special characteristics of a multi-label stream simultaneously, such as label dependency concept drift and recurrent concept. Therefore, in order to meet the above challenges, we develop an efficient ensemble scheme for multi-label data streams aiming at taking into account label dependencies as well as dealing with different types of concept drift.

## 3. Proposed Method

### 3.1. Change Detection Based on Jensen–Shannon Divergence

Most of the previous change detections [31–33] focus on detecting change in the error-rate of the classifiers, without considering changes in the distribution of data. Nevertheless, we adopt the two-window scheme, which is also the most extensively used method by comparing the data distribution between two consecutive windows.

Let $W_1$ and $W_2$ denote reference window and current window, respectively. As shown in Figure 2, the size of both the windows is $n$. The problem of change detection in data streams is to decide the null hypothesis $H_0$ against alternative hypothesis $H_1$ as below:

$$\begin{cases} H_0 & d(W_1, W_2) \leq \varepsilon \\ H_1 & d(W_1, W_2) > \varepsilon \end{cases}$$

where $d(W_1, W_2)$ is a distance function that measures the dissimilarity of two windows and $\varepsilon$ is a distance-based threshold used to decide whether a change occurs. A change occurs if the dissimilarity measure between two windows exceeds a given threshold.

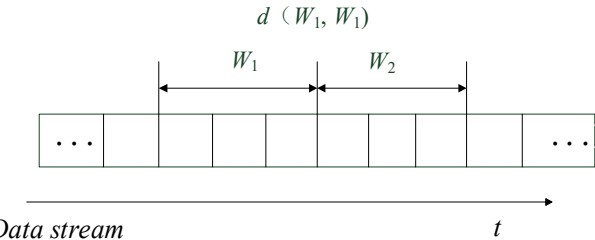

**Figure 2.** Two-window change detection model.

Kullback–Leibler divergence, which is called the relative entropy, is the most popular and robust metric for distance between two distributions [34]. For two discrete distributions *P* and *Q*, there are two probability functions $p(x)$ and $q(x)$, and Kullback–Leibler divergence is defined as

$$KL(P\|Q) = \sum_{x\in\chi} p(x)log\frac{p(x)}{q(x)}$$

where *x* is the space of events. This measure is always non-negative and is zero if and only if the distributions are identical. However, it is not symmetric and does not satisfy the triangle inequality.

To overcome this defect, Jensen–Shannon divergence is adopted as the measure defined in Equation (2), which is a symmetric and bounded variant of the Kullback–Leibler divergence.

$$JS(P\|Q) = \sum_{x\in\chi}(p(x)log\frac{2p(x)}{p(x)+q(x)} + q(x)log\frac{2q(x)}{p(x)+q(x)}) \tag{2}$$

**Theorem 2.** *Let $W_1$ and $W_2$ denote two consecutive windows, where $W_2$ contains the most recent instances. With probability $1 - \delta$, we have $|\mu_{W1} - \mu_{W2}| \le 2\varepsilon$, where $\varepsilon$ is the Hoeffding bound and $\mu_{W1}$ and $\mu_{W2}$ are the mean of two windows [35].*

In general, three types of concept drifts can be detected: sudden, gradual, and recurring concept drifts. The input of the change detection consists of a data stream *S* and window size *n*. The output is an alarm if a change occurs. At the initial stage, the reference is initialized with the first batch of data streams. The current window moves on the data streams and captures the next batch of data stream. When a change is detected, the change detector makes an alarm. If a change does not occur, the basic window $W_2$ slides step by step until the change is detected. A concept drift is detected when the measure exceeds a threshold and a recurring concept is recognized when the measure results in a value of zero. Meanwhile, a more general data-driven approach called bootstrap is adopted to get the threshold. The pseudo-code of the concept drift detection is presented in Algorithm 1.

---

**Algorithm 1 Two-windows-based change detection.**

---

**input:** data streams S, window size $n$;
**output:** ChangeAlarm;
01: **Initialization** $t = 0$;
02: Set reference window $W_1 = \{x_{t+1}, \ldots, x_{t+n}\}$;
03: Set current window $W_2 = \{x_{t+n+1}, \ldots, x_{t+2n}\}$;
04:        **for** each instance in $S$ **do**
05:                calculate $d(W_1, W_2)$ according to Equation (2);
06:                calculate $\varepsilon$ according to Theorem 2;
07:                    **if** $d(W_1, W_2) > \varepsilon$ **then**
08:                            $t \leftarrow$ current time;
09:                            Report a change alarm at time $t$;
10:                            Clear all the windows and goto 02;
11:                        **else if** $d(W_1, W_2) == 0$ **then**
12:                                alarm a recurring concept;
13:                            **end if**
14:                **else** slide $W_2$ by 1 point;
15:                    **end if**
16:            **end for**
17: **end**

---

*3.2. Ensemble Classifier for Multi-Label Data Streams Using Change Detection*

3.2.1. The Framework of Our Method

Ensembles are popular approaches for enhancing classification accuracy in the single-label learning environment. However, they must be generalized for data streams environments. Change detectors are often a part of online classifiers that ensure quick reactions to sudden changes. Unfortunately, they are not capable of detecting gradual changes; whereas, block-based algorithms can outperform online approaches in a slowly drifting environment.

Based on these findings, an efficient ensemble with internal drift detection was proposed. The framework of our method integrates change detection technique and ensemble learning for data streams. The internal drift detector is employed to decide when to build and add new component classifiers.

We first give the general framework described in Figure 3, and then describe the particular model and its analysis. In this paper, a series of component classifiers are built from different portions of the data stream, which maintains a pool of classifiers (each classifier represents one of the existing concepts) and predicts the class of incoming instances using a weighted voting weighted scheme. Furthermore, a change detector was used to monitor the distribution between two consecutive windows, which represent the older and the more recent instances, respectively. Once a change occurs, a new classifier is learned and then a new concept is identified and added to the pool. Meanwhile, the repository of stored historical concepts is checked for reuse.

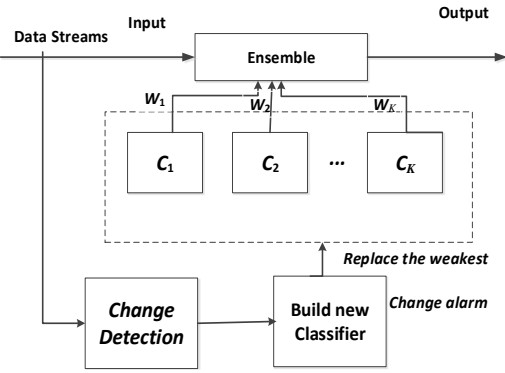

**Figure 3.** The framework of our method.

### 3.2.2. Exploiting Label Dependency

For exploiting dependencies among the labels, the algorithm uses pruned sets (PS) method pruning away infrequently occurring label sets from the training data. A frequent label combination is the one whose occurring number is not less than $m$ ($m > 0$) times of an average value. The $f(c)$ denotes the occurring number of a label combination $c$ in the whole data stream.

$$f(c) = m\frac{n}{2^{|L|} - 1} \tag{3}$$

where $|L|$ represents the size of the label set and $2^{|L|}-1$ is the total number of all label combinations. Then $n/(2^{(|L|)} - 1)$ denotes the average occurring number of all label combinations.

### 3.2.3. Weighting Mechanism

It uses weighted ensemble classifiers to handle concept drifting. The idea is to train an ensemble or group of classifiers from sequential chunks of the data stream. The weights of classifiers are updated dynamically based on their performance on current data. Only the top $k$-classifiers are kept. The final decision is made by the weighted majority votes of the classifiers.

For each incoming instance $x$, each ensemble member $C_i \in E$ is weighted according to Equation (4) when concept drift is detected.

$$w_i = \frac{1}{MSE_r + MSE_i + \varepsilon} \tag{4}$$

A very small positive value $\varepsilon$ is used to avoid division by zero. $MSE_r$ is the mean square error of a randomly predicting classifier, which is used as a reference point to the current class distribution.

$$MSE_r = \sum_y p(\mathrm{y})(1 - p(y))^2$$

$MSE_i$ is the prediction error of classifier $C_i$ on the instances of the most recent window $W$. $f_y^i(x)$ denotes the probability given by classifier $C_i$ that $x$ is an instance of class $y$.

$$MSE_i = \frac{1}{|W|} \sum_{(x,y)\in W} \left(1 - f_y^i(x)\right)^2$$

At the same time, a new classifier $C'$ is built on the instances in $W_2$, weighted, and added to the ensemble. If the ensemble is full, the weakest classifier, i.e., the component with the lowest weight, is replaced by new one based on the result of the evaluation.

In our experiments, we will use Multi-label Hoeffding Trees as component classifiers, but one could use any online learning algorithm as a component learner. The PS is employed at each leaf node of a Hoeffding Tree. The pseudo-code of MLAW is listed in Algorithm 2.

---

**Algorithm 2 MLAW Algorithm.**

---

**input:** *S*: multi-label data streams, *k*: number of ensemble members
**output:** *E*: ensemble of *k* weighted classifiers
01: **begin**
02:   $E \leftarrow \emptyset$;
03:   **for** *all instances $x_t \in S$* **do**
04:     $W \leftarrow W \cup \{x_t\}$;
05:   **if** change detected == true **then**
06:       create a new classifier $C'$;
07:         update the weight of all classifiers in ensemble;
08:     **if** $|E| < k$ **then** $E \leftarrow E \cup C'$;
09:           **else** prune the worst classifier;
10:     **else if** concept is recurring
11:         reuse the classifier in *E*;
12:             **end if**
13:           **end if**
14:         **end if**
15:   **end for**
16: **end**

---

## 4. Experiments

The experiments were performed on 3.0 GHz Pentium PC machines with 16 GB of memory, running Microsoft Windows 8. The experiments were implemented in Java with the help of Massive Online Analysis (MOA). MOA is a software environment for implementing algorithms and running experiments for online learning [36]. Software is available at http://moa.cms.waikato.ac.nz/.

### 4.1. Datasets

The multi-label datasets are from different domains, which can be found in the Mulan website (http://mulan.sourceforge.net/datasets-mlc.html). The datasets are described in Table 1. |D|: the number of distances, |X|: the number of attributes (b: binary/n: numeric), |L|: the number of possible labels; and n and d indicate numeric and binary attributes, respectively.

**Table 1.** The characteristics of datasets.

| Dataset | Domain | |D| | |X| | |L| | LC |
|---|---|---|---|---|---|
| TMC2007-500 | Text | 28,596 | 500b | 22 | 2.219 |
| Medical Mill | Video | 43,907 | 120n | 101 | 4.376 |
| IMDB | Image | 95,424 | 1001b | 28 | 1.999 |
| Random Tree | Tree-based | 100,000 | 30b | 6 | 1.8 → 3.0 |
| RBF | RBF-based | 100,000 | 80n | 25 | 1.5 → 3.5 |

Label Cardinality (*LC*) is a standard measure showing how multi-label is a dataset and it is simply the average number of labels associated with each instance.

$$LC = \frac{1}{|D|} \sum_{i=1}^{|D|} |y_i| \qquad (5)$$

We adopted three real-world datasets: TMC2007-500, Medical Mill, and IMDB. We generated the datasets into stream by MOA. The synthetic dataset Random Tree and Radial Basis Function (RBF) were generated by MOA. Random Tree consists of 100,000 instances with sudden concept drift. RBF dataset contains of 100,000 instances with gradual concept drift.

*4.2. Evaluation Metrics*

This paper introduces four popular performance metrics specifically designed for multi-label data streams classifications: *Hamming loss*, *Subset accuracy*, *F1*, and *Log-Loss*.

(1) *Hamming loss*: The *Hamming loss* averages the standard 0/1 classification error over the m labels and hence corresponds to the proportion of labels whose relevance is incorrectly predicted. *Hamming loss* is defined as Equation (6):

$$Hamming\ \text{loss} = \frac{1}{N} \sum_{i=1}^{N} \frac{y_i \oplus \hat{y_i}}{m} \tag{6}$$

where $\oplus$ represents the symmetric difference between two sets. When considering the Hamming loss as the performance measure, the smaller the value, the better the algorithm performance is. For the next measures, greater values indicate better performance.

(2) *Subset accuracy* is defined in [12], on a window of N instances, as:

$$Subset\ accuracy = \frac{1}{N} \sum_{i=1}^{N} I(y_i = \hat{y_i}) \tag{7}$$

(3) *F1* measure can be described as a weighted average of the recall and precision measures. The calculation equation is as follows:

$$F1 = \frac{2 \times Recall \times Precision}{Recall + Precision} \tag{8}$$

(4) *Log-Loss* distincts from other measures because it punishes worse errors more harshly, and thus provides a good contrast to other measures [20].

$$Log - Loss = \frac{1}{NL} \sum_{i=1}^{N} \sum_{j=1}^{L} \min(log - \text{loss}(\hat{y}_j^{(i)}, y_j^{(i)}), \ln(N)) \tag{9}$$

where

$$\log - \text{loss}(\ \hat{y}, y) = -(\ln(\hat{y})y + \ln(1 - \hat{y})(1 - y))$$

*4.3. Methods and Results*

4.3.1. Parameter Sensitiveness

The experimental design consisted of two independent variables: size of ensemble and the size of window. Firstly, the size of ensemble *k* was varied from 5 to 20 (*k* = 5, 7, 10, 12, 15, 18, 20) to see how it affects the performance of the algorithms. Figure 4 presents the accuracy of MLAW and MLOzaBag on TMC2007-500 while using different sizes of ensemble. Each curve demonstrates the relationship between the size of the ensemble and the subset accuracy of the classification.

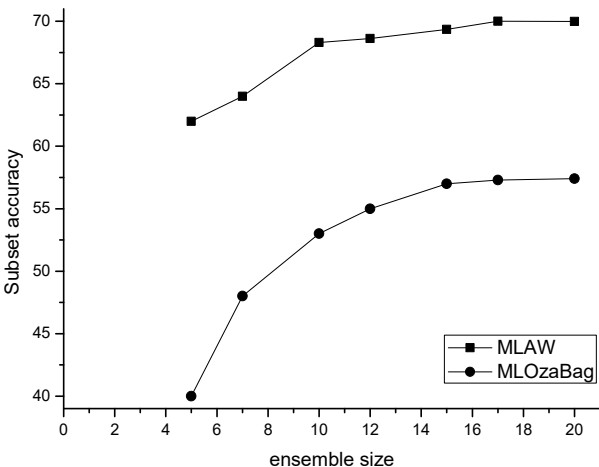

**Figure 4.** Varying the size of ensemble.

MLOzaBag is an ensemble method bagging for *k* models. Compare to MLOzaBag, MLAW is not quite sensitive to the size of ensemble. The accuracy increases when the ensemble has more base classifiers. As there is no strong dependency upon the size of ensemble in terms of accuracy, we chose *k* = 10 as the value in the comparison experiments.

Next, we fix *k* to 10 and further analyze the effect of the size of the window on the performance of our algorithm. The experimental design consisted of the independent variable: window size. The size of window *w* was varied from 500 to 2000, in steps of 250. Tables 2 and 3 present the *Hamming loss* and *Subset accuracy* of our method on five datasets while using different sizes of windows.

**Table 2.** *Hamming loss* using different sizes of windows.

| | Sizes of Windows | | | | | | |
|---|---|---|---|---|---|---|---|
| | **500** | **750** | **1000** | **1250** | **1500** | **1750** | **2000** |
| TMC2007-500 | 0.0524 | **0.0498** | 0.0579 | 0.0603 | 0.0599 | 0.0625 | 0.694 |
| Medical Mill | 0.0132 | 0.0126 | **0.0109** | 0.0218 | 0.0179 | 0.0230 | 0.0289 |
| IMDB | 0.0993 | 0.0963 | 0.0986 | **0.0850** | 0.0911 | 0.0979 | 0.0985 |
| Random Tree | 0.2030 | 0.1183 | **0.1035** | 0.1891 | 0.2100 | 0.2301 | 0.2354 |
| RBF | 0.0753 | 0.0698 | 0.0650 | **0.0642** | 0.0653 | 0.0668 | 0.0697 |

**Table 3.** *Subset accuracy* using different sizes of windows.

| | Sizes of Windows | | | | | | |
|---|---|---|---|---|---|---|---|
| | **500** | **750** | **1000** | **1250** | **1500** | **1750** | **2000** |
| TMC2007-500 | 71.69 | **72.31** | 72.23 | 72.01 | 71.61 | 72.01 | 71.25 |
| Medical Mill | 76.68 | 77.59 | **78.69** | 78.12 | 78.02 | 77.12 | 77.01 |
| IMDB | 81.16 | 81.25 | **81.36** | 81.54 | 80.95 | 81.53 | 82.01 |
| Random Tree | 93.16 | **93.30** | 93.26 | 93.10 | 93.01 | 93.05 | 90.00 |
| RBF | 88.58 | 87.90 | **89.89** | 89.05 | 89.49 | 89.71 | 89.69 |

We observe that if the window size is too small, it will not provide enough amounts of data for a new classifier to be accurate. If a window size is too large, it may contain more than one concept, making the adaption to new concepts slowly. MLAW is not quite sensitive to the window size. As there is no strong dependency upon the window size in terms of *Subset accuracy* and *Hamming loss*, we set *w* = 1000 in the comparison experiments.

### 4.3.2. Comparative Performance Study

This part demonstrated the experimental results regarding the effectiveness and efficiency of the proposed method. The proposed algorithm was evaluated against three algorithms: ML-SAM-*k*nn, Multi-label Hoeffding Tree (MHT), and a bagged of Multi-label Hoeffding Tree (MLOzaBag).

The methods are evaluated using prequential evaluation method with a window of $w = 1000$ instances. All the algorithms use the default parameters. We chose Multi-label Hoeffding Trees as the base classifier. For all the ensembles, we used $k = 10$. The performance were evaluated in terms of *Hamming loss*, *Subset accuracy*, *F*1, and *Log-Loss*, as shown in Tables 4–7 (the best results for each dataset are indicated in bold).

**Table 4.** *Hamming loss* of different algorithms.

|  | ML-SAM-*k*nn | MHT | MLOzaBag | MLAW |
|---|---|---|---|---|
| TMC2007-500 | 0.0475 (2) | 0.1706 (4) | 0.0521 (3) | **0.0509 (1)** |
| Medical Mill | 0.1013 (3) | 0.0665 (2) | **0.0392 (1)** | 0.1039 (4) |
| IMDB | **0.0456 (1)** | 0.1095 (3) | 0.1304 (4) | 0.0986 (2) |
| Random Tree | 0.0541 (2) | 0.0614 (4) | 0.0589 (3) | **0.0135 (1)** |
| RBF | 0.0811 (2) | 0.1400 (4) | 0.0934 (3) | **0.0650 (1)** |
| Average Rank | 2.00 | 3.40 | 2.80 | 1.80 |

**Table 5.** *Subset accuracy* of different algorithms.

|  | ML-SAM-*k*nn | MHT | MLOzBag | MLAW |
|---|---|---|---|---|
| TMC2007-500 | 0.213 (3) | 0.170 (4) | 0.521 (2) | **0.673 (1)** |
| Medical Mill | 0.197 (4) | 0.265 (3) | **0.392 (1)** | 0.301 (2) |
| IMDB | 0.154 (2) | **0.195 (1)** | 0.134 (3) | 0.129 (4) |
| Random Tree | 0.550 (4) | 0.574 (3) | 0.689 (2) | **0.792 (1)** |
| RBF | 0.678 (3) | 0.590 (4) | **0.746 (1)** | 0.703 (2) |
| Average Rank | 3.20 | 3.00 | 1.80 | 2.00 |

**Table 6.** *Log-Loss* of different algorithms.

|  | ML-SAM-*k*nn | MHT | MLOzBag | MLAW |
|---|---|---|---|---|
| TMC2007-500 | 7.9 (3) | 7.5 (2) | 8.2 (4) | **6.2 (1)** |
| Medical Mill | 15.2 (3) | **9.8 (1)** | 19.8 (4) | 13.0 (2) |
| IMDB | **5.3 (1)** | 19.0 (4) | 7.1 (3) | 9.3 (2) |
| Random Tree | 6.4 (2) | 17.5 (4) | 7.6 (3) | **3.8 (1)** |
| RBF | 3.7 (2) | 5.8 (4) | 4.6 (3) | **2.7 (1)** |
| Average Rank | 2.20 | 3.00 | 3.40 | 1.40 |

**Table 7.** *F*1 of different algorithms.

|  | ML-SAM-*k*nn | MHT | MLOzBag | MLAW |
|---|---|---|---|---|
| TMC2007-500 | 0.214 (3) | 0.026 (4) | 0.285 (2) | **0.457 (1)** |
| Medical Mill | 0.028 (4) | 0.034 (3) | 0.042 (2) | **0.103 (1)** |
| IMDB | **0.105 (1)** | 0.021 (4) | 0.087 (2) | 0.016 (4) |
| Random Tree | 0.173 (3) | 0.103 (4) | **0.261 (1)** | 0.254 (2) |
| RBF | 0.389 (2) | 0.274 (4) | 0.378 (3) | **0.436 (1)** |
| Average Rank | 2.60 | 3.80 | 2.00 | 1.80 |

As seen, MLAW outperformed all the other methods under most measures of predictive performance, particularly under *Subset accuracy*, *F*1, and *Log-Loss*. Meanwhile, MLAW achieved the overall best performance in most datasets except IMDB. This is partly due to the management of the recurrent change detection mechanism, which is capable of reusing previous concepts and gaining

better performance in different situations, particularly under concept drift environments. When a new frequent label combination is recognized, the learning model will be updated by the instances with this new frequent label combination, and then the updated learning model can efficiently make multi-label classification before concept drifts take place.

Table 8 provides us with the running time (training + testing) of the compared algorithms. It can be observed that ML-SAM-*k*nn consumed the least, followed by MHT, and MLOzaBag consumed the most time. These results clearly demonstrate the realization of the recurring concept of repeated reuse; hence, our method consumes less time than MLOzBag.

**Table 8.** Running time of different algorithms (CPU second).

|  | **ML-SAM-*k*nn** | **MHT** | **MLOzBag** | **MLAW** |
|---|---|---|---|---|
| TMC2007-500 | **8 (1)** | 14 (2) | 45 (4) | 36 (3) |
| Medical Mill | **34 (1)** | 71 (2) | 457 (4) | 89 (3) |
| IMDB | **285 (1)** | 680 (4) | 359 (2) | 543 (3) |
| Random Tree | 38 (3) | 19 (4) | 87 (2) | **9 (1)** |
| RBF | **113 (1)** | 133 (2) | 201 (4) | 152 (3) |
| Average Rank | 1.40 | 2.40 | 3.20 | 2.60 |

To summarize, single classifier like MHT and ML-SAM-*k*nn performed worse than the other ensemble methods, particularly under datasets with concept drift. The reason is that ensemble classifiers adopt a combination of models to obtain a better predictive performance than the single model. MHT and ML-SAM-*k*nn lack a detection mechanism and therefore adapt ineffectively to concept drift. These results clearly show that MLAW consistently boosts the performance in different data streams scenarios.

Figure 5 shows the ability of MLAW coping with concept drift. When the sudden drift occurs at the 50*K* time stamps, all of the methods react to the changes with a dynamic drop of plotting subset accuracy except our algorithm. The reason is that the proposed algorithm can promptly detect concept drift, according to changes in the concept, and in a timely manner build a classifier to achieve the deal with this type of concept drift. Hence, MLAW suffered the smallest subset accuracy drops. However, the plotting of MHT and ML-SAM-*k*nn appear to be more fluctuated when compared with MLAW, since they are not suitable for non-stationary data streams environments.

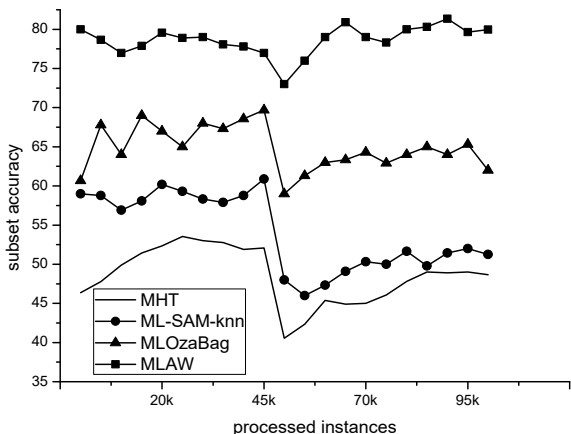

**Figure 5.** Subset accuracy on Random Tree.

## 5. Conclusions

In this paper, we focus on classifying data streams under multi-label scenarios, and a novel ensemble paradigm is proposed. The ensemble proves to be more efficient compared with others and it aims to take label dependency into account and at the same time deal with concept drift.

The experimental results on both synthetic and real-world data streams show that MLAW is capable of constructing a satisfactory model for handling different types of concept drift.

In our future work, we plan to investigate the possibility of adapting the proposed algorithm for tackling concept drift in partially labeled data streams.

**Author Contributions:** Y.S. wrote the manuscript. H.S. commented on the manuscript. S.W. collected and analyzed the references.

**Funding:** This work was supported by the National Natural Science Foundation of China (No. 61672086, No. 61572417, and No. 61702550), the Innovation Team Support Plan of University Science and Technology of Henan Province (No. 19IRTSTHN014), and the Excellent Young Teachers Program of XYNU (No. 2016GGJS-08).

**Conflicts of Interest:** The authors declare no conflict of interest.

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
