# Peer review of "Efficient Ensemble Classification for Multi-Label Data Streams with Concept Drift"

_information, doi:10.3390/info10050158_

Round 1

Reviewer 1 Report

The paper should contain the comparison of experiment results with those achived by other authors - best from literature

Some noticed typing errors:
page 2 line 68 - word missing
page 2 line 80 - let .. denote
page 4 line 173 - replace \delta by \eps
page 4 line 179- two
page 5 line 191 - correct grammar
page 7 Alg 2 - 04 current, 17 calc.ensemble
page 8 line 271 - correct grammar
page 8 line 284- for multi-label classification what is meant by tp, tn, fp, fn?

;

Author Response

Thank you for your letter and for the reviewers’ comments concerning our manuscript. Those comments are all valuable and very helpful for revising and improving our paper, as well as the important guiding significance to our researches. We have studied comments carefully and have made correction which we hope meet with approval. Revised portions are marked in red in the paper. The main corrections in the paper and the responds to the reviewer’s comments are as following.

Point 1:

The paper should contain the comparison of experiment results with those achieved by other authors-best from literature.

Response 1:  According to your suggestion, we supplement the experiment in Section 4 to compare MLAW to state-of-the-art approach ML-SAM-knn. Moreover, we have rewritten the description of experiment and evaluation to give a clearer and correct explanation of them. The modification is marked in red. Please read them carefully, and give your valuable suggestions. Thanks.

Point 2:

Some noticed typing errors:

page 2 line 68 - word missing

page 2 line 80 - let .. denote

page 4 line 173 - replace \delta by \eps

page 4 line 179- two

page 5 line 191 - correct grammar

page 7 Alg 2 - 04 current, 17 calc.ensemble

page 8 line 271 - correct grammar

page 8 line 284- for multi-label classification what is meant by tp, tn, fp, fn?

Response 2:

In the revised manuscript, we correct the sentences. This modification is marked in red. Thanks.

Other Changes

A teacher with overseas study experience has been invited to check our manuscript. In terms of his suggestions, we corrected many syntax errors in our paper, which are marked in red. Please read them. Please read them carefully, and give your valuable suggestions. Thanks.

We appreciate for Editors/Reviewers’ warm work earnestly, and hope that the correction will meet with approval.

Once again, thank you very much for your comments and suggestions.

Reviewer 2 Report

Dear authors,

Enclosed you will find the comments about the proposed work.

Author Response

Response to Reviewer 2 Comments

Thank you for your letter and for the reviewers’ comments concerning our manuscript. Those comments are all valuable and very helpful for revising and improving our paper, as well as the important guiding significance to our researches. We have studied comments carefully and have made correction which we hope meet with approval. Revised portions are marked in red in the paper. The main corrections in the paper and the responds to the reviewer’s comments are as following.

Point 1:

The authors should present their motivation and/or rationale behind the weighted voting methodology?

Response 1:  According to your suggestion, we have rewritten the description of motivation in Section 3.2.2 to give a clearer and correct explanation of them. This modification is marked in red. Please read them carefully, and give your valuable suggestion. Thanks.

Point 2:

Please replace Lines 226 with

(3)

where

Response 2:

In the revised manuscript, we correct the sentence in Section 3.2.2. This modification is marked in red. Thanks.

Point 3:

In Line 298, the authors refer that k was varied from 5 to 20. They should note

EXACTLY what values were used for parameter k. From Figure 4, I can assume that the

algorithm was evaluated using k = 5,7,10,12,15,17,20.

Response 3:

In the revised manuscript, we correct the sentence in Section 4.3.1. This modification is marked in red. Thanks.

Point 4:

Why the authors have selected these datasets? Are there any similar works in the literatures which have utilized the same datasets?

Response 4:

In the revised manuscript, we have added the literatures which have utilized the same datasets. This modification (in Section 4.1) is marked in red. Please read them carefully, and give your valuable suggestions. Thanks.

Point 5:

The presentation of the manuscript should be significantly improved. I suggest the authors to utilize LateX to improve the presentation of the all equations. In some cases the reader is hard to follow. Additionally, the authors should enlarge all figures.

Response 5:

In the revised manuscript, we have redrawn all figures. This modification is marked in red. Please read them carefully, and give your valuable suggestions. Thanks.

Point 6:

In Line 253, the authors refer “In these experiments, we use three real world datasets

and two artificial datasets”. For completeness, they should refer which are the real world datasets and which are the artificial datasets.

Response 6:

This modification (in Section 4.1) is marked in red. Please read them carefully, and give your valuable suggestions. Thanks.

Point 7:

Please replace “The size of window w was varied from 500 to 2000” with “The size of

window w was varied from 500 to 2000, in steps of 250”.

Response 7:

In the revised manuscript, we correct the sentence in Section 4.3.1. This modification is marked in red. Thanks.

Point 8:

Extensive editing of English language is required. There are also many typos. For

example in Line 282: Replace “F1 measure” with “F1-measure”.

Response 8:

In the revised manuscript, we correct the sentence in Section 4.2. This modification is marked in red. Thanks.

Other Changes

A teacher with overseas study experience has been invited to check our manuscript. In terms of his suggestions, we corrected many syntax errors in our paper, which are marked in red. Please read them. Please read them carefully, and give your valuable suggestions. Thanks.

 We appreciate for Editors/Reviewers’ warm work earnestly, and hope that the correction will meet with approval.

Once again, thank you very much for your comments and suggestions.

Round 2

Reviewer 1 Report

Correct Algorithm 2 - input MAXWINDOWSIZE is never used

Author Response

Thank you for your letter and for the reviewers’ comments concerning our manuscript. Those comments are all valuable and very helpful for revising and improving our paper, as well as the important guiding significance to our researches. We have studied comments carefully and have made correction which we hope meet with approval. Revised portions are marked in red in the paper. The main corrections in the paper and the responds to the reviewer’s comments are as following.

Point 1:

Correct Algorithm 2 - input MAXWINDOWSIZE is never used. 

Response Point 1:

In the revised manuscript, we correct the sentences. This modification is marked in red. Thanks.

Reviewer 2 Report

Dear authors,

Enclosed are my comments about the proposed work.

Author Response

Thank you for your letter and for the reviewers’ comments concerning our manuscript. Those comments are all valuable and very helpful for revising and improving our paper, as well as the important guiding significance to our researches. We have studied comments carefully and have made correction which we hope meet with approval. Revised portions are marked in red in the paper. The main corrections in the paper and the responds to the reviewer’s comments are as following.

Response 1:  According to your suggestion, we have rewritten all the equations with Word’s equation editor. This modification is marked in red. Please read them carefully, and give your valuable suggestion. Thanks.

Response 2:

In the revised manuscript, we added the recent citations. This modification is marked in red. Thanks.

Response 3:

In the revised manuscript, we rewritten the Eq. (3) to give a clearer and correct explanation of them.. This modification is marked in red. Thanks.

Response 4:

In the revised manuscript, we correct the sentences. This modification is marked in red. Thanks.

We appreciate for Editors/Reviewers’ warm work earnestly, and hope that the correction will meet with approval.

Once again, thank you very much for your comments and suggestions.

Round 3

Reviewer 2 Report

The authors addressed the previous comments.

I recommend the acceptance of the manuscirpt